# Made in the Womb: Maternal Programming of Offspring Cardiovascular Function by an Obesogenic Womb

**DOI:** 10.3390/metabo13070845

**Published:** 2023-07-13

**Authors:** Mariana S. Diniz, Luís F. Grilo, Carolina Tocantins, Inês Falcão-Pires, Susana P. Pereira

**Affiliations:** 1CNC—Center for Neuroscience and Cell Biology, CIBB—Centre for Innovative Biomedicine and Biotechnology, University of Coimbra, 3004-531 Coimbra, Portugal; mdiniz@cnc.uc.pt (M.S.D.); luis.grilo@uc.pt (L.F.G.); ctsantos@cnc.uc.pt (C.T.); 2Ph.D. Programme in Experimental Biology and Biomedicine (PDBEB), Institute for Interdisciplinary Research (IIIUC), University of Coimbra, 3004-531 Coimbra, Portugal; 3UnIC@RISE, Department of Surgery and Physiology, Faculty of Medicine, University of Porto, 4099-002 Porto, Portugal; ipires@med.up.pt; 4Laboratory of Metabolism and Exercise (LaMetEx), Research Centre in Physical Activity, Health and Leisure (CIAFEL), Laboratory for Integrative and Translational Research in Population Health (ITR), Faculty of Sports, University of Porto, 4200-450 Porto, Portugal

**Keywords:** cardiovascular disease, fetal programming, maternal obesity, maternal physical exercise, mitochondrial function, offspring cardiometabolic remodeling

## Abstract

Obesity incidence has been increasing at an alarming rate, especially in women of reproductive age. It is estimated that 50% of pregnancies occur in overweight or obese women. It has been described that maternal obesity (MO) predisposes the offspring to an increased risk of developing many chronic diseases in an early stage of life, including obesity, type 2 diabetes, and cardiovascular disease (CVD). CVD is the main cause of death worldwide among men and women, and it is manifested in a sex-divergent way. Maternal nutrition and MO during gestation could prompt CVD development in the offspring through adaptations of the offspring’s cardiovascular system in the womb, including cardiac epigenetic and persistent metabolic programming of signaling pathways and modulation of mitochondrial metabolic function. Currently, despite diet supplementation, effective therapeutical solutions to prevent the deleterious cardiac offspring function programming by an obesogenic womb are lacking. In this review, we discuss the mechanisms by which an obesogenic intrauterine environment could program the offspring’s cardiovascular metabolism in a sex-divergent way, with a special focus on cardiac mitochondrial function, and debate possible strategies to implement during MO pregnancy that could ameliorate, revert, or even prevent deleterious effects of MO on the offspring’s cardiovascular system. The impact of maternal physical exercise during an obesogenic pregnancy, nutritional interventions, and supplementation on offspring’s cardiac metabolism are discussed, highlighting changes that may be favorable to MO offspring’s cardiovascular health, which might result in the attenuation or even prevention of the development of CVD in MO offspring. The objectives of this manuscript are to comprehensively examine the various aspects of MO during pregnancy and explore the underlying mechanisms that contribute to an increased CVD risk in the offspring. We review the current literature on MO and its impact on the offspring’s cardiometabolic health. Furthermore, we discuss the potential long-term consequences for the offspring. Understanding the multifaceted effects of MO on the offspring’s health is crucial for healthcare providers, researchers, and policymakers to develop effective strategies for prevention and intervention to improve care.

## 1. Introduction

Obesity is defined as abnormal or excessive fat accumulation in the body, affecting one’s health. The increasing incidence of obesity among women of childbearing age (18–39 years) has made it one of the most common and severe obstetric conditions. Between 2013 and 2014, 37% of women of reproductive age were obese in the United States [1]. Maternal obesity (MO) results in short- and long-term adverse outcomes for the offspring, including an increased risk for cardiovascular disease (CVD) development [2,3]. CVD is the number one cause of mortality worldwide. In 2019, 17.9 million people died from CVD. This represents a massive 32% of the global population. By 2030, it is estimated that more than 23.6 million people will die of CVD [4].

Increasing evidence suggests that MO is a key determinant of offspring’s health not only in the womb but throughout the entire lifetime. Neonates born to obese mothers have an increased predisposition to overgrow, resulting in macrosomia (birth weight, BW > 4000 g) [5]. Moreover, MO offspring have a higher risk of developing congenital anomalies, such as heart defects [6] and neural tube defects [7], increasing the risk of injury during birth, stillbirth, and perinatal death. Later in life, offspring from obese mothers are more prone to develop childhood obesity and chronic diseases [8]. Studies have shown stronger associations between maternal weight gain status, obesity, and increased offspring cardiometabolic risk [9]. A cohort showed increased systolic blood pressure in 6-year-old children of overweight or obese mothers, a primer for higher cardiometabolic risk, emphasizing the link between MO and offspring’s early CVD development [10]. In an Australian cohort, abnormal gestational weight gain was associated with higher systolic blood pressure in 21-year-old offspring [11]. A meta-analysis revealed that human fetuses born to obese mothers exhibited traces of reduced systolic blood pressure and that children (1–12 years old) presented thicker intraventricular septum [12]. These observations strengthen the association between MO and offspring’s cardiovascular health impairment, enhancing the risk of developing CVD in later life [13,14] and challenging the concept that cardiovascular risk is uniquely determined by genetic predisposition and postnatal lifestyle [15]. Despite better healthcare systems, CVD-related events have increased by approximately 50% in children and young adults in recent decades [16]. Although several epidemiological studies explore the relationship between low birth weight and increased risk for CVD development, it is recognized that CVD incidence occurs at both ends of the birth weight spectrum, in a U-shaped curve behavior [17]. In light of the developmental origins of the health and disease (DOHaD) paradigm, the increased number of registered premature deaths caused by CVD could be related to the abrupt rise in MO and malnutrition in the womb.

The currently available epidemiological data on CVD development in offspring of MO are primarily derived from follow-up or meta-analysis studies, and unfortunately, mainly extracted from small population groups. Given this limitation, we strongly advocate for increased attention from epidemiological surveillance entities to define and provide quantified risk assessments of CVD in offspring born to obese mothers. Such efforts are necessary to uncover the implications and establish a comprehensive understanding of the global risk profile. Despite the limitations, efforts have been made to define the prevalence of CVD in offspring born to obese mothers. For instance, in Sweden, offspring born to obese class I (30–34.9 kg/m^2^) mothers have a rate of 0.67/10,000 child-years of developing CVD, while those born to obese class II (35–39.9 kg/m^2^) mothers have a rate of 1.02, and those born to obese class III (>40 kg/m^2^) mothers have a rate of 1.38 [18]. These rates were evaluated using hazard ratios, which were 1.16, 1.84, and 2.51, respectively. Furthermore, other studies have demonstrated a clear association between MO and adverse cardiovascular outcomes in offspring later in life [19]. An epidemiological follow-up study led by Reynolds in 2013 in the UK showed that MO human offspring between the ages of 36 and 62 years old were more likely to be admitted to the hospital due to cardiovascular events and had a higher risk of premature death than the offspring of lean mothers [20]. Nevertheless, it is important to reiterate that more comprehensive and representative data are needed to fully understand the relationship between MO and offspring’s cardiovascular health. Therefore, it has become imperative to understand the mechanisms underlying MO and the offspring’s development of CVD earlier in life and define ways to prevent it. The long-term adverse effects on the offspring’s cardiac function are partially caused by biochemical, structural, morphological, and metabolic function adaptations of the offspring’s cardiovascular system in response to induced stress in utero, imprinting the memory for cardiac dysfunction [21]. These early small changes, maintained over the life course, might result in an ever-growing metabolic derangement and disease. Studying the influence of maternal lifestyle on the offspring’s cardiac function is a noteworthy starting point to explore fetal programming of health and disease. By shedding light on the complex relationship between MO and offspring health, this review aims to provide a deeper understanding of the issue and inform healthcare professionals and researchers about the importance of early identification, prevention, and management strategies. Ultimately, this knowledge can contribute to the development of targeted interventions and policies aimed at improving maternal and fetal outcomes in the context of obesity.

## 2. The Impact of Maternal Obesity on Maternal Health

Maternal obesity has become a significant public health concern worldwide, with its prevalence steadily increasing in recent years. The World Health Organization (WHO) estimates that approximately 20% of pregnant women globally are affected by obesity. This alarming trend is a cause for concern due to the potential adverse effects on both maternal and fetal health. In the context of pregnancy, MO poses specific challenges that can impact not only the well-being of the mother but also the developing fetus.

Pregnancy is a period of physiological adaptations for every maternal organ. Women’s bodies are challenged by pregnancy at metabolic, hemodynamic, hematological, and vascular levels [22]. Therefore, in women with an already compromised physiology, such as obesity, pregnancy may constitute a more drastic challenge and lead to pregnancy-associated diseases, such as gestational diabetes mellitus (GDM), preeclampsia, deterioration of maternal cardiovascular health, and complications during labor and delivery, which can impair maternal and fetal health [21].

Maternal obesity can have profound implications for metabolic regulation during pregnancy due to changes in glucose metabolism to support fetal growth. Due to the increase in fetal glucose usage, higher peripheral consumption, and blood volume expansion, the fasting glucose levels tend to decrease throughout pregnancy [23]. In obese women, maternal blood glucose concentration is higher than in nonobese women [24]. The reduced peripheral insulin sensitivity and insulin rise from the second trimester onwards is physiologic in a lean pregnancy. This effect is mainly driven by the release of hormones with diabetogenic effects, such as cortisol, prolactin, growth hormone, and progesterone [25]. This effect is even more drastic in MO since obese mothers present higher insulin and cortisol blood concentrations at late gestation [26]. The inability of pregnant women to compensate for these metabolic adaptations may result in the development of GDM [27]. Obese pregnant women present lower metabolic flexibility to adapt from fasting to a postprandial state and more postprandial inflammatory markers [28]. Consequently, obese women during pregnancy show a higher risk of developing GDM [29].

Two different phases of lipid metabolism occur during pregnancy: during the first half of the pregnancy, triglycerides are synthesized and stored as energy reserves in maternal adipose tissue; during the second half, lipids are mobilized to the peripheral tissues preparing for lactation [30]. Consequently, circulating triglycerides and cholesterol levels increase during pregnancy in the form of very-low-density lipoprotein (VLDL) and low-density lipoprotein (LDL), while circulating free fatty acids (FA) decrease with pregnancy progression when compared with preconception levels, resulting in a hyperlipidemic environment in the mother—fetus interface [30]. Pregnant obese women show net lipolysis throughout pregnancy in contrast with lean pregnant women who demonstrate anabolic lipogenesis in early gestation and lipolysis in late gestation [31]. Increased maternal circulating triglyceride and fatty acids have been found in MO [32]. Compared with lean women, obese pregnant women show higher monounsaturated and saturated FAs and lower polyunsaturated FAs, higher branched-chain and aromatic amino acids, and increased GlycA levels, a marker of inflammation associated with metabolic dysfunctions, including type 2 diabetes and CVD [33,34,35,36]. Higher levels of branched-chain amino acids have been associated with reduced activity of branched-chain amino acids catabolism enzymes and reduced utilization in liver and adipose tissue causally related to insulin resistance [37].

The entire cardiovascular system, composed of the heart and the closed system of vessels, undergoes physiologic adaptations during pregnancy. Heart rate rises 15–25% via catecholamine release during pregnancy, resulting in 20–30% increased stroke volume and 30–50% increased cardiac output for the first and second trimesters [38,39,40]. Increased myocardial contractility is associated with higher left ventricular (LV) mass, wall thickness, and left atrial diameter compared to nonpregnant women [39,41]. Pregnancy is also a hypercoagulable state, which can predispose the pregnant woman to thromboembolic events [42,43]. Erythropoiesis is stimulated but is not able to compensate for the increase in blood volume, which can lead to anemia [44,45].

Maternal systemic vascular resistance decreases by 10–30% in early pregnancy resulting in lower mean arterial pressure. Consequent peripheral vasodilatation is achieved through the decreased response to vasoconstrictive molecules (e.g., angiotensin II) and the rise of circulating levels of vascular relaxing agents (e.g., nitric oxide, NO) [22,40,46,47]. In overweight and obese women, NO levels are increased in maternal blood at late gestation [48]. Circulating blood leptin and concentration of low-grade inflammatory markers (i.e., C-reactive protein (CRP) and interleukin-6 (IL-6)) are increased in the maternal blood of obese women [49,50]. Together with excessive VLDL, it can lead to endothelial and placental dysfunction, establishing a link between MO, hyperlipidemia, and the development of preeclampsia [51]. Highlighting the role of hyperlipidemia in preeclampsia development, studies in humans have found that mothers who develop preeclampsia present increased peripheral triglyceride levels either before, during, or shortly after pregnancy [52,53]. The highest reported triglyceride concentration was associated with a 4-fold increased risk for preeclampsia development in a case-control study [53]. Consistently, a higher risk of preeclampsia was found in obese women compared with women of normal weight [54].

Subsequent to the lower vascular resistance, the renin–angiotensin–aldosterone system is activated, resulting in water and sodium retention and increased blood volume, which can reach up to 45% before term [55,56]. When the fall in systemic vascular resistance reverses, a low-resistance uteroplacental circulation is created, which helps increase the blood flow to the placenta to support fetal growth requirements [22]. Abnormal placentation is observed in the first weeks of pregnancy and mostly results from the remodeling of the uterine spiral arteries leading to low vascular resistance and provoking systemic endothelial dysfunction [57,58]. The suboptimal occurrence of these processes provokes high-resistance uteroplacental circulation and may result in preeclampsia [59,60]. The mother—fetus interface is critical to the proper fetal development and maternal health; thus, it is critical to understand the impact of MO on the fetoplacental unit to better comprehend its adverse effects on fetal and maternal health.

## 3. Maternal Obesity-Derived Effects in the Fetoplacental Unit

The placenta establishes the interaction between the fetus and the mother, mediating the relationship between fetal development and maternal health through the regulation of nutrient transport, endocrine regulation, and oxygen delivery [61,62]. The placenta is derived from extraembryonic tissues. The precursor of trophoblasts, which forms the outer layer of the blastocyst, is the trophectoderm, which subsequently differentiates into cytotrophoblasts, which then differentiate either into extravillous trophoblasts or syncytiotrophoblasts. Vasculogenesis and hematopoiesis take place, arising placental macrophages, endothelial cells, and vascular smooth cells after the formation of the villous core [63]. The villous core contains the fetal blood vessels covered by the syncytiotrophoblast that contacts the maternal blood in the spiral arteries from the intervillous space [64]. To increase uteroplacental blood flow, placental extravillous trophoblasts invade spiral arteries replacing endothelial cells and degrading smooth muscle, which is critical in normal gestation [65,66,67]. Throughout gestation, the number of cellular layers between fetal and maternal blood decreases, reducing to only syncytiotrophoblast and the fetal capillary endothelium at term [61]. The syncytiotrophoblast is the main solute (e.g., amino acids, glucose) barrier from maternal to fetal blood and is composed of two polarized membranes, the microvillous plasma membrane (MVM) facing maternal blood in the intervillous space and the basal plasma membrane (BM) to fetal capillaries side. The syncytiotrophoblast plasma membranes contain different amounts of transporter proteins regulating the exchange of macro- and micronutrients from maternal to fetal blood [68]. The placenta presents the ability to modulate its function according to the environment to provide the best conditions for fetal growth. This plasticity can be forced and lost in MO due to metabolic, endocrine, and inflammatory dysregulation. Maternal saturated fatty acid and carbohydrate consumption could lead to a state of cardiac glucolipotoxicity, whereas increased levels of maternal glucose and an excess of circulating lipids lead to lipid accumulation as triglycerides in the placenta [13]. Overexposure to macronutrients during fetal development can modulate fetal development and precipitate fetal growth, affecting organ development in a tissue-dependent way. In MO, the placental transport of micronutrients is also affected. In baboons, it has been suggested that compared with lean pregnancies, the placental transcripts of chromatin-bound vitamin D receptor (VDR), vitamin D transporter LRP2, and vitamin D 25-hydroxylase are downregulated, affecting vitamin D placental metabolism and likely leading to depression of the placental vitamin D system [69]. The apparent MO-induced decreased transport of vitamin D may lead to decreased sterols, vitamin-binding proteins, hormones, and lipoproteins. This becomes highly relevant due to the adverse effects of vitamin D deficiency in the offspring: asthma, obesity, and cardiometabolic disease. In addition, a maternal high-fat diet (HFD) composed of an increased ratio of linoleic acid (LA) to alpha-linolenic acid (ALA) and increased total fat content in rats led to an upregulation of genes involved in cholesterol uptake and transport (*ApoA2*, *ApoC2*, *Cubn*, *Fgg*, *Rbp4*, and *Ttr*) in the late-stage placenta in comparison with a maternal HDF composed of a decreased ratio of LA to ALA [70], highlighting that maternal fat dietary content and composition influences MO-induced placental nutrient transport. In fact, placental function and structure are impacted by MO. MO results in an immature placental villous tree (i.e., reduced number of larger diameter villi and abnormal angiogenesis) and hyperplasia of the muscular tissue of placental vessel walls [71,72]. Normal pregnancy is characterized by inflammation in the placenta; however, in MO placentas, an exacerbation of the inflammatory state concomitant with macrophage infiltration is observed [72,73]. Placental nutrient and hormone exchange functions in MO are also affected due to the differential expression of transport protein in both MVM and BM. Fetal nutrient overexposure, such as fatty acids, amino acids, or glucose, occurs in MO and leads to fetal overgrowth due to both the high level of nutrients and fetal insulin hypersecretion [74,75]. Moreover, endocrine receptors are highly expressed in MVM, namely, for insulin, adiponectin, leptin, and insulin-like growth factor-1 (IGF-1) [76,77,78]. These factors are increased in circulation for MO and contribute to altered syncytiotrophoblast function due to downstream activation of signaling pathways, such as mTOR and AMPK, which are key nutrient-sensing pathways [79,80].

Decreased AMPK activation was reported in MO placentas, suggesting lower intracellular ATP levels, which were also reported in MO placentas [80,81]. mTOR plays a critical role in regulating the placental nutritional support to fetal growth by coordinating amino acid transport, folate transport, promoting protein synthesis, and mitochondrial biogenesis and function [82,83,84,85,86,87]. Even though these studies suggest increased mTOR signaling in the placenta from MO, placental mTOR gene expression is reduced in obese women despite upregulated gene expression of proteins associated with mitochondrial function and oxidative stress [88]. However, isolated trophoblasts from MO show reduced mitochondrial density and function [89,90]. High levels of fatty acid transporters in MVM, highly oxidative metabolism, and mitochondrial density in the trophoblast have been reported, suggesting a mitochondrial-dependent metabolism [91,92]. Triglycerides uptake from maternal blood undergoes hydrolysis in MVM, rising fatty acids availability in the placenta [93]. In addition, glutamine has been suggested as a main substrate for oxidative phosphorylation in trophoblasts via the tricarboxylic (TCA) cycle [94]. Gene expression of fatty acid transporters is increased in the MO ovine placentas [95]. Moreover, glycolysis upregulation does not compensate for decreased oxidative phosphorylation and β-oxidation in MO [81,89,96]. Nevertheless, metabolic compensation for fatty acid oxidation by increased peroxisomal activity in MO trophoblast has been reported [89]. Hitherto, as this counterbalance is not total, the placental lipid oversupply results in intracellular lipid storage and accumulation and consequent higher transport to fetal blood [89,97]. Indeed, increased triglyceride content has been found in the placentas of obese women at term, resulting in a lipotoxic environment [97,98].

Consistently, placental liver X receptor (LXR) signaling pathway transcripts are found to be upregulated in obese mothers [99]. In addition, CGI-58 and PLIN2 protein expression is increased in the placenta from MO [98,100]. Oleic acid upregulates CGI-58 and PLIN2 in cultured primary human trophoblast cells regulating lipid turnover in the placenta [100]. Palmitic acid stimulates proinflammatory cytokines (IL-6, TNF-α, and IL-8) expression via JNK/EGR-1 in trophoblasts in vitro similarly to what is observed in MO placenta [101]. Oleic acid is also able to activate mTOR signaling and amino acid uptake via system A sodium-dependent neutral amino acid transporter (SNAT) [102]. Docosahexaenoic acid decreases amino acid uptake through the inactivation of mTOR signaling. [102]. SNAT activity was decreased in the placenta of obese women [103]. Amino acid transport across the placenta occurs against the concentration gradient through transporters and exchangers in MVM and BM commonly co-transporting extracellular sodium [104,105]. Over 20 tightly regulated amino acid transporters are known, including seven neutral amino acid transporters forming system A and system L which vary in substrate specificity and sodium dependency [106]. Leptin stimulates SNAT via signal transducer and activator of transcription (STAT) 3 activation of JAK-STAT signaling [107]. Hyperleptinemia has been reported in obese pregnant women, which correlates with fetal leptin concentrations and is associated with fetal insulin resistance [24,108,109]. Leptin stimulates lipolysis in the placenta and IL-6 and nitric oxide release in primary trophoblasts [110,111,112].

Placental glucose transport and metabolism are also critical for the placenta’s function and fetal growth as fetal endogenous production is minimal. Impaired glucose metabolism leads to fetal insulin resistance and hyperinsulinemia [62]. The main glucose transporter in MVM and BM is glucose transporter 1 (GLUT-1), which is more densely present in MVM, suggesting that BM is the rate-limiting membrane to transport glucose to fetal blood [113,114]. Maternal glucose metabolism is impaired in obese pregnant women compared with normal BMI women, presenting greater fasting, 1 h, and 2 h glucose levels in oral glucose tolerance tests accompanied by hyperinsulinemia [109]. Transplacental glucose transport has also been shown to be increased in MO with fivefold GLUT-1 protein expression in MVM, and a similar increase has been observed in fetal blood glucose concentration [75,115]. Maternal hyperinsulinemia can also stimulate placental glucose uptake and SNAT, and the placenta appears to not develop insulin resistance, resulting in overactivation of amino acid uptake via mTOR signaling [61,77,116,117,118,119]. Insulin-like-growth factor 1 is increased in maternal circulation [120]. IGF-1 stimulates trophoblast proliferation and placental amino acid and glucose transport through the higher expression of GLUT-1 in BM; it promotes protein and carbohydrate metabolisms and induces fetal growth [121,122,123,124].

In normal gestations, the placenta releases inflammatory cytokines into the maternal circulation. Placenta from obese women also presents an exacerbation of inflammation with higher levels of IL-6, TNF-α, MCP-1, IL-8, and CRP in maternal blood [50,73,125]. IL-6 induces SNAT activity through STAT3 signaling and SNAT2 protein expression and fatty acid uptake [126,127,128]. TNF-α activates phospholipase A2 and amino acid uptake via SNAT activation through p38 MAPK signaling [126,127,129]. It is important to take into consideration that these mechanisms have been identified for the inflammatory molecules individually, and a more physiological combination of inflammatory molecules is needed to understand the impact of the inflammatory milieu in placenta cells. Obese pregnant women also present lower adiponectin levels [50,130,131,132,133]. Adiponectin can regulate maternal metabolism and placental function through decreased amino acid and glucose uptake, SNAT, and glucose transporter gene expression downregulation in trophoblast [130,131,132,133,134,135,136,137,138].

Placental lipotoxicity could lead to oxidative stress by originating species such as lipid peroxides, oxidized lipoproteins, and oxysterols that could exacerbate glucose intolerance, insulin resistance, and cardiomyopathy [13]. The determination of oxidant/antioxidant markers in human MO placentas, such as reactive nitrosative species (RNS), reactive oxygen species (ROS), catalase, superoxide dismutase (SOD), malondialdehyde (MDA), by Malti et al. demonstrated an imbalanced redox status associated with MO [139]. Placental proteomics was enriched in antioxidant capacity proteins of MO, suggesting their dysregulated function [140]. Indeed, decreased antioxidant activity, including glutathione concentrations and SOD activity, was reported in the placenta and trophoblast of MO [97,139]. Increased ROS production in the placenta of MO was also reported [81].ROS can react with nitric oxide and produce peroxynitrite and potentiate nitrosative stress, which has been observed in MO placentas, potentially regulating intracellular signaling [141,142]. Despite these findings, it is essential to mention that in moderate concentrations, ROS and RNS play a vital role in placental formation (trophoblast proliferation and angiogenesis [143]), thus being vital for embryonic and fetal development. Exacerbated concentrations of these free radicals may be harmful since in the first trimester of gestation, the placental antioxidant defenses are low, which could easily lead to a pro-oxidative state, quickly affecting fetal development [144].

Additionally, Wallace et al. demonstrated increased levels of a marker of hypoxia, carbonic anhydrase IX, in the placentas of diet-induced obese mice [145]. MO modulates and creates an adverse intrauterine environment characterized by lipotoxicity, hypoxia, and/or oxidative stress (Figure 1), affecting placental development and normal function. Placental malfunction can severely impact organogenesis, especially heart formation. Investigation in fetal sheep hearts revealed that placental dysfunction led to decreased cardiomyocyte cell cycle activity and differentiation rate, indicative of a more immature myocardium, with potentially fewer cardiomyocytes, which could lead to a dysfunctional heart [146].

## 4. Fetal Programming: Womb-Dictated Disease

### 4.1. Maternal Obesity Impacts Heart Formation and Offspring Cardiac Physiology

After the placenta, the heart is the first organ to acquire function during fetal development [147]. During the early-life stage in utero, heart formation occurs in the embryonic stage, resulting in a four-chambered heart [147]. During cardiac formation, the mesodermal progenitor cells specify and differentiate into nonmuscular and muscular cells, forming the cardiac layers, such as the epicardium, endocardium, the internal lining of the heart, and the myocardium [147,148]. The myocardium is composed of cardiomyocytes and cardiac muscle cells that occupy the major volume of the heart and are responsible for the propagation of electrical impulses, allowing the heart to contract incessantly [149,150,151]. During the human embryonic stage, the mononuclear cardiomyocytes continue to grow, which corresponds to the period of hyperplasia or proliferative growth. This process continues until the perinatal/neonatal stages, after which the extracellular matrix suffers alterations, where the transition from mononuclear to binuclear cardiomyocyte occurs, and cardiomyocyte renewal rates become low [150]. The heart loses its regenerative capacity, and this process is called postnatal hypertrophy. During adulthood, cardiomyocyte hypertrophy occurs and is subdivided into two different types: physiological and pathological. Both types serve as a cardiac adaptative response to a determined stimulus, but the mechanisms by which each occurs, and phenotypes are significantly different [152]. Physiological hypertrophy ensures the heart’s normal function and maintains or even increases muscle contractility capacity in response to, for example, high-intensity physical exercise, or even pregnancy [153]. However, pathological hypertrophy, which is characteristic in CVD patients, is responsible for a gradual loss of cardiac contractility and cardiomyocyte death, where increased collagen levels are verified [152]. Pathological cardiac hypertrophy mechanisms are regulated, in part, by the nuclear factor of activated T-cells (NFAT) [152]. This protein is responsible for cardiac hypertrophy by regulating the expression of hypertrophic genes. NFAT dephosphorylation promotes cytoplasmatic translocation to the nucleus, regulating the expression of one of the hypertrophy biomarkers, brain natriuretic peptide (BNP), inducing cardiac hypertrophy [154]. The protein NFAT is regulated by calcineurin [155]. In sheep, both calcineurin and NFATc3 are increased in the LV of mid-gestation MO fetuses [156]. The cardiac antihypertrophic response is mainly mediated through the NO signaling pathway. The activation of the soluble guanylyl cyclase (sGC) enzyme by NO allows for the conversion of guanosine triphosphate (GTP) into 3′,5′-cyclic guanosine monophosphate (GMP), which stimulates the activation of protein kinase G (PKG) [154]. On the one hand, PKG activates the cyclic AMPKα responsive element-binding protein (CREB), responsible for anti-apoptotic mechanisms, regressing cardiac hypertrophy [157]. On the other hand, PKG is responsible for inhibiting the Ca^2+^-sensitive protein phosphatase calcineurin, inhibiting the translocation of NFAT to the nucleus, and preventing the development of cardiac pathological hypertrophy [155,158]. It has been shown that MO compromises the fetal heart and its structure [12,159]. Thickening of the left ventricular free wall, myocyte hypertrophy, thickened intima walls, impaired vascular function, diastolic and systolic dysfunction, and contractile dysfunction with Ca^2+^ homeostasis disruption were reported in MO mice, sheep, and nonhuman primate MO offspring [160,161,162,163,164,165]. In another study, female rat offspring exposed to MO displayed elevated systolic and diastolic blood pressure at 26 and 51 weeks of age, whereas altered blood pressure was not observed in male offspring [166]. Interestingly, a meta-analysis study showed that MO human offspring’s blood pressure was positively correlated with maternal gestational weight gain [167]. Cardiac physiology abnormalities induced by MO were also sex-specific. Smaller LV relative wall thickness and higher myocardial performance were registered for female offspring of HFD mice dams. Contrastingly, male offspring of the same generation presented higher LV relative wall thickness and no differences in myocardial performance [163]. A meta-analysis of 13 human studies concluded that exposure to diabetes in utero, a common complication of obesity, is associated with increased systolic and diastolic blood pressure in male but not in female offspring [168]. Nonetheless, all these studies support that in response to MO adverse in utero environment, fetal heart remodeling occurs in a sex-different manner. Maternal HFD modulates the expression of crucial blood pressure regulatory factors that influence the cardiac structure, and this remodeling could represent a compensatory mechanism to attempt to preserve cardiac function.

### 4.2. Epigenetic Disruption in Maternal Obesity Progeny

Epigenetic mechanisms have been implied in disease programming since these contribute to defining the adult phenotype [169]. A solid relationship was demonstrated between the disruption of epigenetic factors and fetus adaptations to maternal nutrition/placental environment that could persist later in life [170]. Likewise, Heijmans et al. showed altered DNA methylation patterns in insulin-like growth factor 2 (IGF-2), a maternally inherited gene in individuals exposed to famine prenatally, supporting the theory that early life environmental conditions can lead to epigenetic changes that persist throughout life, showing the apparent involvement of epigenetic dysregulation influenced by maternal nutrition [170,171]. Dysregulation of epigenetic modifications, such as DNA methylation and post-translational histone modification, have been described for MO offspring [169]. A study in Sprague Dawley rat MO offspring performed one and ten days after birth, has gathered evidence of downstream signaling of the Class IIa histone deacetylase (HDAC)—myocyte enhancer factor 2D (MEF2) axis via AMPK, which is responsible for the regulation of genes involved in FA oxidation [172]. Class IIa HDAC is phosphorylated by AMPK, which is activated during times of cardiac metabolic stress to increase energy production. Afterward, class IIa HDAC is exported to the nucleus, with the expression of MEF2-dependent genes that respond to hypertrophy marker genes (atrial and brain natriuretic peptides, ANP and BNP). Increased expression of these genes was found along with increased protein expression of enzymes involved in FA oxidation, suggesting transcriptional reprogramming [172]. In addition, maternal HFD downregulated a subset of cardiac miRNAs involved in transforming growth-factor- β (TGF-β)-mediated heart remodeling [173], highlighting the need to diverge more research into the relationship between MO and offspring’s cardiac miRNA levels. Nonetheless, all these studies suggest altered epigenetics due to in utero conditions. These altered epigenetic mechanisms could lead to memory imprinting for CVD development in offspring and its manifestation in adulthood or even early in life, contributing to the increased CVD incidence in children and young adults in recent decades (Figure 1).

### 4.3. Sexual Dimorphism in Fetal Programming

The cardiometabolic risk and CVD risk factors are sex-specific [174]. It is demonstrated that increased CVD risk could start during early exposure to maternal stress factors, such as MO. The available data implicating maternal diet-induced obesity and offspring outcomes are either male-only cohorts [175] or mixed-sex cohorts, the latter less frequent. Very few studies are focused on exploring the sex differences in cardiac metabolism and even less in in utero programming of offspring’s cardiac disease by maternal habits. However, evidence has shown that maternal stress factors, including MO, have different impacts on the offspring’s cardiophysiologic and metabolic parameters according to sex in murine and nonhuman primate animal models [176]. It is necessary to unravel the sex-specific effects of MO on cardiometabolic risk to understand in utero programming to optimize offspring’s healthcare.

#### 4.3.1. The Role of Epigenetic Mechanisms in Sexual Dimorphism

Epigenetic mechanisms modulate the sex-specific gene expression in response to an adverse intrauterine environment. It is well-established that epigenetic mechanisms control pathological cardiac hypertrophy, gene expression control, and genome stability [169]. A genome-wide study found that one type of DNA methyltransferase (DNMT, KDM4A) is increased for knockout mice with the hypertrophic phenotype with pathological hypertrophy and heart failure [177]. These control the promotor for the expression of NFATc4. This could explain the sex-specific cardiac response since some DNMTs are encoded by the X and Y-chromosomes, which are mainly activated by estrogen [177]. Interestingly, diet-induced MO-associated DNA methylation patterns are more pronounced in female offspring livers than in male ones [14]. Thus, it is possible that offspring gene methylation and/or demethylation profiles induced by MO partly control the sex-specific response. Nevertheless, it seems important to highlight the levels of sex steroid hormones with their influence on the epigenetic response and the potential role in CVD sex-specific response in MO offspring.

#### 4.3.2. Sex Steroid Hormones

Sex steroid hormones play a vital role in the organism. These are already expressed in an early embryonic stage. Sex hormones can regulate gene expression and may play a role in predisposition to disease development. Sex differences in physiology and metabolism are important factors that likely define the differences between rates of cardiometabolic disease risk among men and women [174]. Estrogens are prime sex hormones that play an important role in protein modification, gene regulation, and cellular process modulation. It has been shown that mitochondria are important targets of estrogen. Indeed, estrogen receptors α and β are present in mitochondria and can affect mitochondrial bioenergetics and anti-apoptotic signaling [178]. In addition, the sex steroid hormone 17β-estradiol (E2) positively influences cholesterol and lipid metabolism. Estrogen plays an essential role in the cardioprotective effect [179]. Connexin 43 (Cx43) is regulated by E2. Cx43 is a gap junction protein that is highly expressed in human cardiomyocytes and is responsible for muscle synchronized contraction [180] and cardioprotection. Mitochondrial Cx43 is present in subsarcolemmal mitochondria in the cardiac muscle [180]. In addition to interacting with other proteins that are essential for mitochondrial function, Cx43 regulates mitochondrial respiration, oxygen consumption, Ca^2+^ homeostasis, and K^+^ fluxes [181]. Ischemia-reperfusion (I/R) injury occurs when the blood returns to the heart after a lower oxygen supply provoked by myocardial ischemia [182]. Heart mitochondria play an essential role in determining the severity of the I/R injury since these organelles are responsible for providing energy to support cardiac contractility during a few seconds and display the mitochondrial ATP-sensitive potassium channels that play a crucial role in the protection against I/R injury [182]. Due to its action on cardiac mitochondria, Cx43 is essential into regulating recovery in a sex-specific way. Given that E2 regulates Cx43, female mice hearts are more likely to be protected by Cx43: additionally, myocardial Cx43 is more expressed in females than in male rats [183]; on the other hand, male and ovariectomized female Cx43 knockout with posterior E2 administration have shown that E2 mediated-cardioprotection involves Cx43 and is sex-specific [183]. These data suggest that Cx43 could play a role in mitochondrial function sex-specific response not only in response to an I/R injury but, potentially, also to MO-induced cardiac function dysregulation in the offspring, which may justify, in part, the mitochondrial sex-specific responses observed for MO offspring.

## 5. The Metabolic Legacy of an Obesogenic Womb

### 5.1. Maternal Obesity Modulates the Offspring’s Cardiac Insulin Signaling Pathway

Insulin resistance refers to a condition in which the body’s cells become less responsive to the hormone insulin. Insulin is produced by the pancreas and helps regulate glucose levels in the blood. When the cells become insulin resistant, they are less efficient at absorbing glucose from the bloodstream. Initial metabolic malfunctions in glucose oxidation generally occur due to diminished entrance of glucose through GLUT-4 and augmented fatty acid collection through inhibition of glucokinase and pyruvate dehydrogenase. Amplified fatty acid flux into myocardial cells due to systemic tissue insulin resistance and insulin resistance-related reorganization of cluster differentiation protein 36 (CD36) to the plasma membrane are associated with increased fatty acid oxidation [184]. Insulin resistance is a key contributing factor to MO-induced cardiac lipotoxicity. However, the opposite is also true. Cardiac lipotoxicity could lead to insulin resistance since toxic FA metabolites have the ability to impair myocardial insulin signaling [13]. Specifically, the cardiac tissue of 8-week-old MO offspring mice showed cardiac hypertrophy associated with hyperinsulinemia and activation of insulin signaling pathways through increased phosphorylated Akt-1 (Ser473) and mTOR (Thr 2448) [160]. These changes could potentially lead to LV reconstruction at an accelerated level, which could, in turn, lead to heart failure [160]. Moreover, in the cardiomyocytes of MO offspring mice, decreased Akt phosphorylation and AMPK activity and phosphorylation of insulin receptor substrate 1 (IRS-1) in Ser307 were shown to be significantly enhanced [185]. Elevated phosphorylation of IRS-1 in Ser307 in hearts from fetuses born to obese sheep were also established along with impaired insulin sensitivity and myocardial dysfunction [13]. In a maternal overfeeding ovine model, the activation of the PI3K/Akt/mTOR/ signaling pathway was verified in the fetal and neonatal stages of the progeny [186]. Nonetheless, all this supports that impairment in the cardiac insulin signaling pathway induced by MO (Figure 1) could be a key contributing factor to MO-induced offspring cardiac dysfunction.

### 5.2. Dysregulation of Glucose and Fatty Acid Oxidation in the MO Offspring

The myocardium presents a reduced ability to synthesize FAs; thus, it depends on the uptake of free FAs or lipoproteins to maintain cardiac lipid homeostasis and energy metabolism [187]. In MO late-gestation mice fetuses, the serum lipidic composition is altered, with females exhibiting larger alterations in the FAs composition of phospholipids. An altered lipidome is also verified in the fetal heart, with the promotion of lipid metabolism in a sex-specific way [188]. There is evidence of incomplete β-oxidation of long-chain FAs and altered oxidative metabolites in myocytes differentiated from stem cells derived from the umbilical cord from MO offspring at birth [189]. The described insulin resistance impairs the heart’s ability to effectively use glucose as a source of energy. As a result, the heart is forced to rely even more on FAs as an energy source. However, due to the impaired ability to properly utilize and metabolize these FAs, they accumulate within the heart muscle cells, which can lead to lipotoxicity. In 6-month-old male mice offspring, MO induced increased cardiac palmitoyl carnitine-supported mitochondrial respiration, along with alterations in cardiac carbohydrate and lipid metabolism [165]. Regarding lipotoxicity, enzymes that participate in long-chain FAs oxidation are upregulated to keep up with the high demands of the increased FAs substrates. Nonetheless, this upregulation is not enough to withstand the increased load of FAs through oxidative phosphorylation in obesity, resulting in the accumulation of fatty acyl-CoA in the mitochondria [190]. Moreover, newborn mice offspring cardiomyocytes from HFD and streptozotocin-treated mothers showed a lower response to exogenous palmitate, a substrate for mitochondrial FAs oxidation [162]. Overall, incomplete, or impaired FAs oxidation may lead to cardiac lipotoxicity in the hearts of MO offspring (Figure 1), impairing key energy pathways in MO offspring’s cardiac function.

### 5.3. The Missing Link: Exploring the Role of Mitochondrial Dysfunction in the Development of MO-Induced Cardiovascular Disease in the Offspring

Cellular bioenergetics dysregulation can play a crucial role in the pathophysiology of cardiovascular disease. Mitochondrial function and structure are highly susceptible and responsive to environmental alterations [190]. It has been suggested that mitochondria show developmental plasticity, undergoing adaptations depending on the internal cellular surrounding environment, which supports the DOHaD theory. Therefore, mitochondria could play a pivotal role in developmental programming and disease manifestation later in life [190]. If mitochondria become dysfunctional, at least the production of energy in the cardiomyocyte and the production of cell-specific components needed for normal cell function become unbalanced. An imbalance between energy production and energy demand and a disturbance in energy transfer networks play an important role in various pathologies [191,192]. Mitochondria play pivotal roles in various other cellular processes, including metabolism, Ca^2+^ buffering, oxidative stress regulation, and cell death regulation. Thus, any disruption in mitochondrial function can have detrimental effects on cardiac health and may contribute to the development of cardiovascular disease. We should consider that subtle in utero early mitochondrial adaptations could persist throughout life, creating a dysfunctional feedback loop and causing progressive and even more accentuated metabolic derangement that would culminate in disease. The starting point can be an affected oxidative phosphorylation system (OXPHOS) function, with limited ATP supply and increased generation of ROS and RNS, which can lead to mitochondrial DNA (mtDNA) and cellular components damage, inciting ROS/RNS leak and aggravating mitochondrial dysfunction [190]. Being an organ with high energy demand, the heart requires its fuel to keep an optimal contractile activity. Therefore, changes in cardiac metabolism have been linked to oxidative stress, mitochondrial dysfunction, and mitophagy [162]. Proposed mechanisms for mitochondrial dysfunction include mitochondrial structural adaptations (cristae effacement) [193,194], impairment in electron transport chain (ETC) complex activities, defects in the assembly and organization of OXPHOS supercomplexes, oxidative stress, decreased expression of cardiolipin, impaired TCA cycle anaplerosis, and mitochondrial uncoupling [195].

Cardiac mitochondrial dynamics is a quality control process, vital to mitochondrial health maintenance. An unbalance of fission and fusion events has been linked to cardiometabolic complications, contributing to an increased number of dysfunctional mitochondria. Mdaki et al. showed fragmented and low-membrane potential mitochondria in cardiomyocytes of maternal HFD mice offspring with little or no fusion or fission events, evidencing severe damage that could signal cell apoptosis [162]. Cardiac fusion protein expression was shown to be sex-specific in maternal HFD rat offspring [196]. Male MO offspring presented higher cardiac mitochondrial fusion protein levels (optic atrophy 1, OPA1), along with increased fusion events. Males presented cardiac post-translational modifications, which are known to affect dynamism and influence mitophagy-mediated cell death, suggesting that maternal HFD leads to a sex-divergent response [196]. In another study, where the mothers were given a high fat—high sugar (HFHS) diet, both females and males presented increased levels of phosphorylated mitochondrial fission factor (p-MFF), linked to mitochondrial fission, suggesting altered cardiac mitochondrial dynamics in MO offspring [163]. However, it was also described that offspring born to mothers given an HFHS presented decreased cardiac transcript levels of the mitochondrial fusion protein (MFN-1), associated with mitochondrial fusion, in comparison with the control [196]. Since mitochondrial dynamics are fundamental to cardiac development, maturation, and function, impairment of these events could lead to developmental programming of the cardiac tissue, leaving cardiometabolic imprinting and impacting cardiac health at birth and throughout the lifetime.

Mitochondrial dysfunction is often characterized by an impairment in ETC complexes’ efficiency and a reduction in their coupling with ATP synthesis. Three-month-old maternal HFHS diet male mice offspring showed reduced cardiac respiration associated with complex I, complex II, and complex II + III [163]. An abnormal expression of genes involved in mitochondrial dynamics decreased mitochondrial ETC complex activity and expression and citrate synthase activities; decreased mitochondrial respiration affected cardiac energy production, impacting the offspring’s cardiac function [163].

Oxidative and nitrosative stress are implicated in several diseases. A pro-oxidative cellular state has been pointed out as one of the significant indicators for obstetric and fetal complications. From the intrauterine environment and placenta, through cardiac development, to the adult stage, findings have implicated oxidative stress in the origin and perpetuation of CVD in MO offspring [175]. In fetal hearts of mothers fed a HFD, transcript levels of SOD2 and hypoxia-inducible factor α (HIF-1α) were reduced [197]. Consistently, reduced SOD2 expression was observed in the aortas of 6-month-old rat offspring of obese dams [198]. In 7-day-old mice offspring exposed to MO, cardiac SOD and reduced glutathione (GSH) and glutathione peroxidase (GPx) activities were decreased both in males and in females [199]. Interestingly, the LV tissue of mice offspring born to mothers given an HFHS diet presented increased levels of SOD2 in comparison with the control [195]. This is possibly in response to oxidative damage (especially lipid peroxidation) through increased levels of MDA in these offspring [195]. In fact, MDA was increased in cardiomyocytes of newborn Sprague Dawley MO rats [162]. During MO, lipid peroxidation can lead to calcium overload and increased mitochondrial uncoupling, causing cardiac dysfunction [200]. Thus, the maternal effects of MO, indeed, affect the offspring postnatally, pointing to mitochondria-mediated oxidative stress as a mechanism for the development of CVD in MO offspring (Figure 1).

## 6. Intervention Strategies for Addressing the Metabolic Derangements Caused by Maternal Obesity

To counteract the adverse effects of MO on the offspring’s cardiometabolic health, several studies suggested positive impacts of maternal lifestyle interventions on the offspring’s cardiometabolic function and gut microbiota, including physical activity, nutritional adaptations, and supplementation [171,172]. Exercise before and during gestation in combination with an adequate and balanced diet resulted in lower male offspring’s blood glucose concentration, insulin levels, and plasma triglyceride levels, and alterations in gut microbiota diversity, such as enriched operational taxonomic units (OTUs) affiliated with Bacteroides and Blautia [201]. The UK Pregnancies Better Eating and Activity Trial (UPBEAT), consisting of an 8-week intensive lifestyle intervention in human offspring’s diet and physical activity, prevented the concentric cardiac remodeling observed in 3-year-old offspring of obese women subjected to standard antenatal care, including the increased interventricular septum, posterior wall, and relative wall thickness [202].

Other interventions have been studied in animals and humans—for example, research investigating the role of omega fatty acid supplementation during pregnancy [203,204,205,206]. Conjugated linoleic acid supplementation in HFD-fed Sprague Dawley rats partially improved endothelial function in 150-day-old offspring fed a chow diet after weaning [205]. Fish oil (FO) supplementation improved insulin sensitivity and cardiac function in 70-day-old Sprague Dawley offspring of HFD-fed dams but had detrimental effects on the offspring of chow diet-fed mothers [205]. Similar results were observed in 100-day-old male offspring of HFD-fed mothers, but no alterations were reported in the offspring of mothers fed with the FO-supplemented control diet [204]. Pregnant women (BMI ≥ 25 kg/m^2^) were given FO from recruitment to 3 months postpartum. Despite lower maternal and offspring triglycerides levels, the offspring’s body composition at two weeks of age and insulin resistance three months postpartum were unaltered by FO supplementation [207]. The long-term effects of this study are yet to be evaluated.

Dysbiosis is observed during MO and is associated with poorer offspring health [208]. As such, the incorporation of prebiotics, probiotics, and natural extracts in the maternal diet has been explored to prevent metabolic derangements in MO offspring. Six-month-old C57BL/6J mice offspring of MO mothers supplemented with the prebiotic polydextrose during pregnancy showed decreased adipose tissue mass and improved glucose tolerance compared with offspring of MO but not supplemented mothers [209]. After challenging the offspring with an obesogenic diet, polydextrose prevented the accelerated weight gain and resulted in gut microbiota changes with increased Bacteroides cells in fecal samples, compared with offspring of control mothers [209]. Supplementation with *Elateriospermum tapos* yogurt during HFD-induced MO pregnancy in Sprague Dawley rats reduced the levels of cardiometabolic parameters (triglycerides, cholesterol, LDL, leptin, and liver enzymes) in the plasma of 21-day-old offspring [210].

As pro-oxidative environment-induced responses are implicated in MO offspring programming, the role of synthetic and natural antioxidants, such as N-acetyl cysteine (NAC) and curcumin, respectively, and their incorporation in the maternal diet during pregnancy has been explored [199,211]. Supplementation of NAC in C57/B6 pregnant MO mice induced through western diet (WD) and sucrose attenuated the left ventricular hypertrophy in 7-day-old MO-offspring, decreasing heart weight, collagen content, and interventricular septal thickness, and prevented the overexpression of genes of the NADPH oxidases (NOX) enzyme complex (e.g., Ncf2, Cyba, Rac1), especially in males. NAC supplementation also prevented the alterations in the expression of proteins involved in the Akt-mTOR signaling (e.g., Akt, phosphorylated Akt at Ser 473, phospho-S6) [199].

Several nutritional and supplementation interventions have been studied in the context of MO offspring metabolic programming (Figure 2). Despite some studies using the same strategy (e.g., supplementation) and similar models, the supplement dose, intervention period, evaluated tissues, postnatal age, and metabolic parameters highly differ, resulting in variable outcomes. Even though supplementation is advised in some cases (i.e., disease and nutrient deficiencies), health guidelines still recommend whole-food interventions in detriment to supplements because nutrient sufficiency and variety can be guaranteed only by diet adjustments in healthy individuals [212]. In addition, supplements can only be taken for a determined period; otherwise, they might induce adverse outcomes in an individual’s health [213]. This could represent a major problem, given that in some areas, foods rich in macro- and micronutrients may not be affordable for the whole population. Addressing these barriers is required to further promote healthier maternal dietary patterns. In addition to diet supplementation and whole-food enhancement, other strategies can be implemented during MO to improve both the mother’s and the offspring’s cardiometabolic health. Exercise has long been associated with a general improvement in metabolic health and is recommended during gestation. Exercise is potentially an effective noninvasive strategy to improve the offspring’s mitochondrial health. However, the type of exercise and the molecular benefits of physical training for the offspring’s cardiovascular health remain highly unexplored.

### 6.1. The Influence of Maternal Physical Exercise during Gestation on Offspring’s Overall Health

Regular exercise has been linked to improved cardiovascular system function in nonpregnant obese individuals [214]. Improved metabolic adaptations, such as increased FAs oxidation, were observed in the hearts of exercised female Sprague Dawley rats [215]. Moreover, mitochondrial biogenesis was increased in the cardiac tissue of exercised male B6129SF2/J mice [216], possibly due to AMPK enhancement and subsequent activation of mitochondrial peroxisome proliferator-activated receptor gamma coactivator 1-alpha (PGC1α). In addition, an increased ability of cardiac mitochondria to oxidize FAs was verified in exercised mice, increasing the capacity for ATP synthesis [217]. Whole-body insulin sensitivity was also improved in human individuals who regularly exercised, with increased Akt activation and membrane translocation of GLUT4 in the skeletal muscle [218]. The benefits of maternal physical exercise on offspring’s health have been discussed in recent years [219]. Maternal physical exercise prevented the rise in triglycerides and reduced leptin and fat mass in MO offspring albino Wistar rats [220]. Maternal physical exercise also improved the expression of hepatic mitochondrial genes involved in mitochondrial biogenesis, FAs metabolism, and TCA cycle activity in 8 months-old male Sprague Dawley offspring of HFD mothers [221]. Impaired NO metabolic pathway during pregnancy could increase the risk of offspring CVD development since a deficit in NO bioavailability induced by MO may persist in the postnatal stage through adulthood [222,223], with possible adverse cardiovascular outcomes. Maternal physical exercise during an obesogenic pregnancy increased NO derivates (nitrites and nitrates) serum levels in 14-week-old male offspring fed the WD compared with offspring born to sedentary mothers [175,224]. The increased levels of nitric oxide derivates induced by maternal exercise may be the origin of an improved cardiovascular function of the offspring.

### 6.2. Enhancing MO Offspring Cardiovascular Health through Maternal Physical Exercise

The impacts of maternal physical exercise on the offspring’s vasculature or cardiac function have not been extensively studied. As mentioned in Section 3, the fetoplacental unit is essential to fetal development and is highly affected by MO. Vascular function was improved in 17-week-old male C57BL/6N mice offspring of voluntarily exercised mothers fed a WD, with increased NO levels in the plasma and reduced endothelial expression of the Sod1 transcript [175]. However, the positive effect of maternal exercise was dependent on the offspring’s diet, only observed in WD-fed offspring [176], denoting an important role of voluntary maternal exercise when offspring are second-challenged in the postnatal period but potentially an insufficient effect of voluntary exercise when the offspring maintained a standard chow diet after weaning. A study involving a diet-induced MO in C57BL/6 mice showed that 8-week-old male offspring of obese but exercised mothers had decreased cardiac hypertrophy and improved ejection fraction compared with MO offspring [225]. Maternal physical exercise prevented the increased heart weight, cardiomyocyte cell area, atrial natriuretic peptide (Nppa) transcript, and Myh7/Myh6 ratio in the offspring’s cardiac tissue while contributing to increased expression of Troponin I, activated Troponin I (phosphorylated in the residues Ser 23/24), Tropomyosin, and sarco/endoplasmic reticulum Ca^2+^-ATPase (SERCA2) proteins [225].

A human study including newborns of obese women exercised from the 14th week of gestation until delivery revealed no impact of exercise on the obesity-induced cardiac effects observed by echocardiography in 1–3-day-old and 6–8-week-old babies (decreased global strain, strain rate, Doppler velocities, and the thicker intraventricular septum) [226]. Of note, the exercise protocol consisted of 3-day-per-week supervised training and 1-day-per-week home exercise, and the maternal compliance to physical exercise was 1.3 ± 0.8 and 0.8 ± 0.7 sessions per week, respectively, which could have contributed to the ineffectiveness of the exercise intervention. Given the positive influence evidenced by human milk oligosaccharides for the breastfed offspring, the effect of maternal physical exercise on the offspring’s cardiometabolic health focused on the role of the 3′Sialyllactose (3′SL) breastmilk oligosaccharide has been explored [227]. In the study by Harris et al, 52 weeks-old C57BL/6 mice offspring of HFD-fed mothers were cross-fostered between sedentary (SED) and voluntarily exercised (TRAIN) dams. Male, but not female, SED-TRAIN-fostered offspring had enhanced cardiac ejection fraction and consequently improved cardiac function, compared with TRAIN-SED [227]. Maternal physical exercise was responsible for the increased 3′SL content in the breastmilk of humans, which was positively correlated with average activity at two months postpartum, regardless of BMI. HFD-fed exercised mice had increased 3′SL concentration in the breastmilk, compared with HFD mice, which had reduced 3′SL levels [227].

Efficient strategies to improve MO offspring outcomes, especially regarding the offspring’s cardiovascular health, need further research and uniformization between studies. The beneficial effects of diet supplementation are highly dependent on several factors, including the type of diet the offspring eat after weaning, sometimes showing advantages only when the offspring are second-challenged (Figure 2). Although maternal physical exercise seems to improve cardiovascular health in animal models, it is necessary to consider that translation of these studies implicates the mothers’ compliance with exercise protocols. Nevertheless, more studies are needed to fully comprehend MO offspring programming mechanisms for disease and the beneficial role of maternal exercise during MO pregnancies.

## 7. Final Remarks

This review collected evidence that maternal in utero obesogenic environment and fetoplacental dysfunction play a significant role in programming the cardiac metabolic function of the offspring, leading to a modulation of the offspring’s risk for the development of CVD in a sex-dependent manner. Epigenetic alterations have been identified as potential factors for the long-term imprinting of CVD risk in offspring exposed to obesogenic metabolic overload during pregnancy. These alterations may contribute to the modulation of gene expression patterns and metabolic programming in the offspring. Moreover, lipotoxicity, insulin resistance, impaired mitochondrial dynamics, dysfunctional OXPHOS activity, and oxidative/nitrosative stress were identified in animal models for MO offspring as consequences of MO exposure. Further research is necessary to fully understand the effects of MO on the cardiovascular function of the offspring, particularly on MO offspring’s cardiac mitochondrial function. Alongside the research on cardiac mitochondrial function, exploring novel diagnostic, preventative, and therapeutic strategies can be crucial to preventing CVD in offspring exposed to MO during pregnancy. Maternal physical exercise emerges as a potential intervention that could be implemented to mitigate the adverse effects of MO and improve cardiovascular health outcomes in the offspring. In summary, this review highlights the importance of maternal in utero environment, fetoplacental dysfunction, and epigenetic alterations in programming the cardiac metabolic function of the offspring and influencing their risk for CVD development. The identified consequences of MO exposure on the offspring’s cardiac health underscore the need for further investigation, particularly in understanding the impact on mitochondrial function. Implementing strategies such as maternal physical exercise may offer promising avenues for prevention and intervention to mitigate the risk of CVD in offspring exposed to MO during pregnancy.

## Figures and Tables

**Figure 1 metabolites-13-00845-f001:**
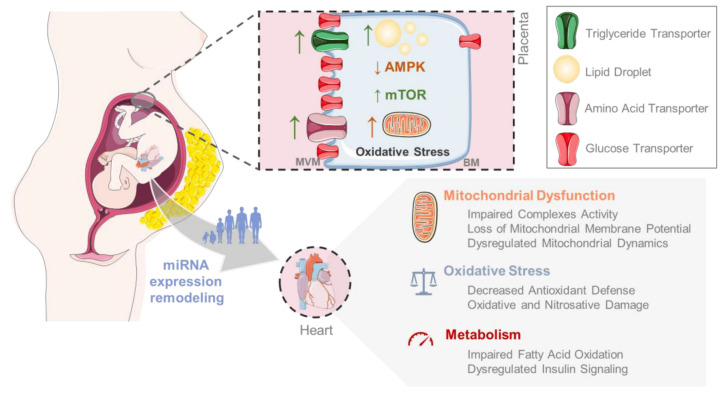
Maternal obesity (MO) can program the offspring to develop cardiovascular disease (CVD) through various mechanisms. MO induces placental dysfunction characterized by increased expression of triglycerides and amino acid transporters, enhanced lipid droplets, decreased AMP-activated protein kinase (AMPK) activation, decreased mTOR gene expression, and the presence of oxidative stress. Moreover, MO contributes to offspring’s cardiac epigenetic and metabolic remodeling, resulting in impaired fatty-acid oxidation, dysregulated insulin signaling, mitochondrial dysfunction, and elevated oxidative stress.

**Figure 2 metabolites-13-00845-f002:**
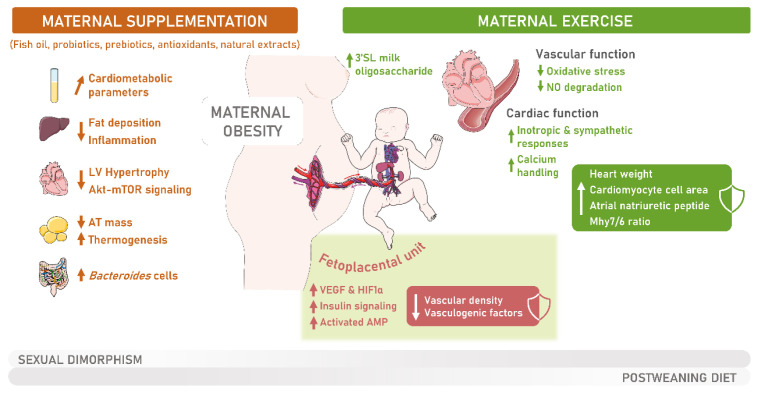
Detailing the beneficial effects of maternal supplementation and physical exercise during maternal obesity (MO) for the offspring’s cardiometabolic and cardiovascular health. Maternal supplementation during pregnancy contributes to the improvement of cardiometabolic parameters in the plasma of MO offspring (e.g., triglycerides, cholesterol, low-density lipoproteins, leptin), reduced liver fat deposition and inflammation, cardiac protein kinase B (Akt)—mammalian target of rapamycin (mTOR) signaling and left ventricle (LV) hypertrophy, increased thermogenesis in the adipose tissue (AT), with reduced AT mass and increased Bacteroides cell levels in MO offspring fecal samples. The beneficial effects of maternal physical exercise for the offspring’s cardiovascular health include changes in the fetoplacental unit, where it contributes to increased levels of the vascular endothelial growth factor (VEGF), hypoxia-inducible factor 1 alpha (HIF1α), activated AMP-activated protein kinase (AMPK) and downstream insulin signaling while preventing the decreased vascular density and vasculogenic factors levels induced by MO. Exercise ameliorates MO offspring vascular function through diminished endothelial oxidative stress and nitric oxide (NO) degradation and enhances the offspring’s inotropic and sympathetic responses and calcium handling in the heart while protecting the offspring against increased heart weight, cardiomyocyte cell area, atrial natriuretic peptide, and cardiac alpha (α)-myosin heavy chain (Mhy)7/Mhy6 ratio, contributing to an overall improved cardiovascular health. In addition, maternal exercise promotes increased levels of breastmilk oligosaccharide 3′ Siallylactose (3′SL), which potentially improves MO offspring’s metabolic and cardiac function. Of note, the effects observed due to maternal supplementation or physical exercise during pregnancy highly depend on the offspring’s sex and type of diet after weaning.

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
