# Peer review of "Made in the Womb: Maternal Programming of Offspring Cardiovascular Function by an Obesogenic Womb"

_metabolites, 2023, doi:10.3390/metabo13070845_

Round 1

Reviewer 1 Report

1.     In Line 161-162: Together with excessive VLDL, it can lead to endothelial and placental dysfunction, establishing a link between MO, hyperlipidemia, and the development of preeclampsia [49].

a.     Please add more detail to this link. This is a key concept and needs more attention

2.     These lines 179-182 need clarification: The fetal  trophectoderm origins the human placenta and differentiates into trophoblast, and extraembryonic mesoderm later developing into fibroblasts, endothelial cells, and macro- phages in the villous core develop.

3.     This is a very important point about macronutrients Line 209-211: Fetal nutrient overexposure, such as fatty acids, amino acids, or glucose, occurs in MO and leads to fetal overgrowth both due to the high level of nutrients and via fetal insulin hypersecretion [67,68].

a.     What I would like to see added is a section on what is known or lacking about the transfer of micronutrients – minerals, fat soluble vitamins, etc. Omega three and six related compounds, for example, impact inflammation and intake is variable. Antioxidants also may have altered transfer and would play a role in fetal and infant growth and outcome. In caring for these infants, they are overgrown, and the quality of their growth is poor. Is there evidence that there is dysregulation of other important components of tissue growth?

4.     Excellent discussion of sex associated differences in cardiac outcomes.

5.     Interesting discussion of mitochondrial dysfunction in the cardiac cells. I believe there is emerging evidence of mitochondrial dysfunction in placental cells as well in preeclampsia. This likely has an impact on delivered nutrients. Is this associated with MO?

6.     Great to see this point (Lines 612-13) included here in section 6: I would like to have seen mechanism discussed earlier in the paper as noted in point 3. Other interventions have been studied in animals and humans. For example, research investigating the role of omega-fatty acid supplementation during pregnancy [198–201].

7.     Figures are very clear. In Figure 2 you focus on supplementation as opposed to whole food diet enhancement. Please address the advantages to whole foods as compared to supplementations, and the barriers to diet alterations at a population level.

Minor editing will be needed, the use of prepositions could be improved

Author Response

Coimbra, June 29th, 2023

Reference: Comments to the reviewers for the manuscript metabolites-2455069: “Made in the Womb: Maternal Programming of Offspring Cardiovascular Function by an Obesogenic Womb”

Dear Drs. Susanne Aufreiter & Bo Li,

We are submitting below the comments to the reviewers’ questions on the review manuscript entitled “Made in the Womb: Maternal Programming of Offspring Cardiovascular Function by an Obesogenic Womb” by Diniz et al. (Manuscript ID: metabolites-2455069) to the special issue “Fetal–Maternal–Neonatal Metabolomics” of the Metabolites journal. We are very thankful for the reviewers’ comments which were valuable and certainly helped us refine our manuscript. We have replied to the reviewers’ concerns and revised the manuscript accordingly. The alterations are highlighted in the manuscript in yellow.

Reviewer 1 (R1)

R1 - 1. In Line 161-162: Together with excessive VLDL, it can lead to endothelial and placental dysfunction, establishing a link between MO, hyperlipidemia, and the development of preeclampsia [49]. Please add more detail to this link. This is a key concept and needs more attention.

Authors (A): We highly value and acknowledge the dedication of the Reviewer 1 and greatly appreciate his valuable comments, which have significantly contributed to the improvement of our manuscript. Taking into consideration the insightful suggestions provided by the reviewer, we have made several important revisions. We agree that addressing the link between hyperlipidemia and preeclampsia can be a key concept to improve our manuscript. Accordingly, from lines 174 to 179, we comment on two studies that have shown that maternal hypertriglyceridemia, widely associated with MO, could lead to preeclampsia development.

R1 - 2. These lines 179-182 need clarification: The fetal trophectoderm origins the human placenta and differentiates into trophoblast, and extraembryonic mesoderm later developing into fibroblasts, endothelial cells, and macrophages in the villous core develop.

A: We agree with the reviewer that this sentence wasn’t clear. Thus, we rephrased it from lines 196 to 202.

R1 - 3. This is a very important point about macronutrients Line 209-211: Fetal nutrient overexposure, such as fatty acids, amino acids, or glucose, occurs in MO and leads to fetal overgrowth both due to the high level of nutrients and via fetal insulin hypersecretion [67,68]. What I would like to see added is a section on what is known or lacking about the transfer of micronutrients – minerals, fat soluble vitamins, etc. Omega three and six related compounds, for example, impact inflammation and intake is variable. Antioxidants also may have altered transfer and would play a role in fetal and infant growth and outcome. In caring for these infants, they are overgrown, and the quality of their growth is poor. Is there evidence that there is dysregulation of other important components of tissue growth?

A: We acknowledge that in the initial version of our manuscript, we inadvertently overlooked the significant aspect of micronutrient placental transport. Recognizing the crucial role of micronutrients in fetal growth as signaling molecules (such as retinoic acid), structural components (such as zinc in transcription factors), or catalysts (such as copper), we appreciate the reviewer's insightful suggestions. In response to their feedback, we have incorporated relevant studies (lines 221-235) that shed light on the compromised micronutrient transport across the placenta associated with Maternal Obesity (MO).

R1 - 4. Excellent discussion of sex associated differences in cardiac outcomes.

A: We sincerely thank the reviewer for recognizing and appreciating our comprehensive discussion of sex-associated differences in cardiac outcomes.

R1 -5. Interesting discussion of mitochondrial dysfunction in the cardiac cells. I believe there is emerging evidence of mitochondrial dysfunction in placental cells as well in preeclampsia. This likely has an impact on delivered nutrients. Is this associated with MO?

A: Unfortunately, specific literature linking maternal obesity (MO) and placental mitochondrial dysfunction in preeclampsia is lacking, thus we can only suggest a few mechanisms by which this might occur. MO is a risk factor for preeclampsia development. This is thought to be due to MO-induced structural alterations in the placenta and insulin resistance (10.3389/fphys.2018.01838). Insulin resistance affects mitochondrial function through several mechanisms, mainly through increased production of reactive oxygen species (ROS), prompting oxidative stress. In preeclampsia, increased placental ROS have been identified (10.1016/j.bbadis.2018.12.005), thus it is very likely that preeclampsia-associated oxidative stress could be stimulated by MO-led insulin resistance. The other mechanism by which this can occur is through MO-induced insulin resistance modulation of PGC1-α expression, which is responsible for mitochondrial biogenesis, a highly important process to maintain mitochondrial fitness. In preeclampsia, severe downregulation of PGC1- α has been found, which was associated with mitochondrial dysfunction (10.1016/j.yexcr.2017.07.029). Thus, MO-induced insulin resistance may be a key player in MO-induced placental mitochondrial dysfunction in preeclampsia.

R1 - 6. Great to see this point (Lines 612-13) included here in section 6: I would like to have seen mechanism discussed earlier in the paper as noted in point 3. Other interventions have been studied in animals and humans. For example, research investigating the role of omega-fatty acid supplementation during pregnancy [198–201].

A: We fully acknowledge that the suggestion provided in point 3 has had a significant impact on enhancing the quality of our manuscript. The insightful feedback provided by the reviewer has undoubtedly contributed to the improvement of our work, and we are thankful for their valuable input.

R1 - 7. Figures are very clear. In Figure 2 you focus on supplementation as opposed to whole food diet enhancement. Please address the advantages to whole foods as compared to supplementations, and the barriers to diet alterations at a population level.

A: As per the reviewer's suggestion, we have explicitly discussed the benefits of whole foods over supplementation and examined the challenges associated with implementing dietary modifications at a population level. This valuable input from the reviewer has been incorporated into our manuscript, specifically in lines 683-691, to provide a more comprehensive analysis on the topic.

R1 - 8. Minor editing will be needed, the use of prepositions could be improved.

A: We would like to express our appreciation to the reviewer for bringing this to our attention. We carefully considered this aspect during the revision process of the final version of the manuscript.

Reviewer 2 Report

The authors have provided a highly comprehensive review of the impact of maternal obesity on cardiovascular disease in offspring. Overall, the manuscript is well-written. I have one suggestion for the authors: while this review is thorough, it lacks sufficient epidemiological data and measurements regarding the effect of maternal obesity on the development of cardiovascular disease in offspring. For instance, it would be valuable to include information on the prevalence of cardiovascular disease among offspring of obese mothers compared to those of mothers with a normal BMI. Additionally, including risk ratios or odds ratios for these two groups would further enhance the completeness of the review. If the authors can obtain and incorporate such data, it would significantly strengthen their review.

Author Response

Reviewer 2 (R2)

R2: The authors have provided a highly comprehensive review of the impact of maternal obesity on cardiovascular disease in offspring. Overall, the manuscript is well-written. I have one suggestion for the authors: while this review is thorough, it lacks sufficient epidemiological data and measurements regarding the effect of maternal obesity on the development of cardiovascular disease in offspring. For instance, it would be valuable to include information on the prevalence of cardiovascular disease among offspring of obese mothers compared to those of mothers with a normal BMI. Additionally, including risk ratios or odds ratios for these two groups would further enhance the completeness of the review. If the authors can obtain and incorporate such data, it would significantly strengthen their review.

A: We sincerely appreciate the time and effort the reviewer dedicated to providing valuable comments that greatly contributed to the improvement of our manuscript. We fully agree that incorporating data on increased risk ratios or odds ratios for cardiovascular disease (CVD) in offspring born to obese mothers would significantly reinforce the findings presented in our review. Unfortunately, it is regrettable that there is currently a lack of representative data on this specific risk that accurately reflects the worldwide population. Existing meta-analyses and follow-up studies, at present, only represent small population groups, providing a limited understanding of the global scenario. However, in response to the reviewer's suggestion, we have included a study conducted in Sweden (lines 79-91) that examines the risk of CVD development in offspring born to obese mothers. While this study offers valuable insights, we emphasize in our manuscript that its findings may not be universally applicable and should not be generalized to the global population. Given this limitation, we strongly advocate for increased attention from epidemiological surveillance entities to define and provide quantified risk assessments of CVD in offspring born to obese mothers. Such efforts are necessary to better comprehend the implications and establish a comprehensive understanding of the global risk profile. Once again, we sincerely appreciate the reviewer's valuable input and emphasize the need for further research in this area.

We would like to thank the reviewers for their invaluable feedback. Your insights and suggestions have greatly contributed to the improvement of our manuscript. We sincerely appreciate the time and effort you dedicated to providing constructive comments. Your input has been instrumental in shaping the final version of our work. We are grateful for your valuable contributions and for helping us enhance the quality of our manuscript.

We hope this manuscript is now suitable for publication.

With our best regards,

Susana P. Pereira, PhD

MitoXT - Mitochondrial Toxicology and Experimental Therapeutics Laboratory

CNC - Center for Neuroscience and Cell Biology, University of Coimbra, 

UC Biotech Building, Lot 8A, Biocant Park

3060-197 Cantanhede, Portugal

pereirasusan@gmail.com
